# HyperAdapter: Generating Adapters for Pre-Trained Model-Based Continual Learning

## Abstract

Humans excel at leveraging past experiences to learn new skills, while artificial neural networks suffer from the phenomenon of catastrophic forgetting during sequential learning. Efforts have been made to alleviate forgetting by introducing a rehearsal buffer into the model, but this way is impractical in real-world scenarios with data privacy. Recently, pre-trained model-based continual learning methods have provided new insights into addressing this issue by effectively utilizing the powerful representational capabilities of pre-trained models to avoid catastrophic forgetting without a rehearsal buffer. In this work, we propose a novel pre-trained model-based continual learning framework, HyperAdapter, which utilizes a hyper-network to generate adapters based on the current input, adapting the pre-trained model to the corresponding task. This paradigm requires fewer additional parameters as the number of tasks increases, which is a critical advantage for scaling to long sequences continual learning. Unlike methods that partition task-related knowledge into relatively independent subspaces, it promotes positive knowledge transfer across tasks. Comprehensive experiments across various datasets demonstrate that HyperAdapter consistently outperforms all existing methods and even exceeds the upper bounds of multi-task learning, establishing a new state-of-the-art for pre-trained model-based continual learning. Our code will be released.

## 1 Introduction

Humans exhibit remarkable abilities for constantly acquiring new knowledge in the dynamically changing real world, which helps them grasp new skills more easily with a richer knowledge base. However, when trained on successive task stages, neural networks tend to overfit on the current task and perform poorly on previous ones, a problem known as catastrophic forgetting (McCloskey & Cohen, 1989). Continual learning aims to learn new tasks while retaining past knowledge (De Lange et al., 2021; Mai et al., 2022). Inspired by the replay process in the hippocampus, some approaches rely on a rehearsal buffer to store samples from previous tasks (Buzzega et al., 2020; Cha et al., 2021; Chaudhry et al., 2019). These samples are then combined with data from the current task to mitigate forgetting. While these methods have shown promising results, they face challenges in real-world scenarios with privacy concerns (Shokri & Shmatikov, 2015) or memory constraints (Smith et al., 2021). The need for rehearsal-free methods to address continual learning in practical settings remains.

Without a rehearsal buffer, methods need to focus on the parameters and architecture of the model. Drawing inspiration from synaptic consolidation in the neocortex, EWC (Kirkpatrick et al., 2017) prevents forgetting of critical past knowledge by applying regularization to the model weights. While regularization-based methods (Kirkpatrick et al., 2017; Li & Hoiem, 2017) offer new insights into rehearsal-free continual learning, they alleviate forgetting at the expense of model plasticity, leading to suboptimal performance when learning new tasks. Recent advances in pre-training (Chen et al., 2020; He et al., 2020; Bao et al., 2021; Xie et al., 2022; He et al., 2022a) inspire the community to integrate pre-trained models into continual learning, aiming to prevent forgetting while efficiently learning new tasks. In particular, prompt-based methods (Wang et al., 2022b;a; Smith et al., 2023) can even outperform rehearsal-based methods. These methods maintain a prompt pool for the pre-trained backbone, selecting prompts based on input samples to instruct the learning of corresponding tasks. However, a small pool may lead to forgetting, while a large one burdens memory and hinders positive knowledge transfer between tasks, presenting the stability-plasticity dilemma (Jung et al., 2023). EASE (Zhou et al., 2024) introduces the adapter into continual learning to enhance the adaptability

Figure 1: Motivation of HyperAdapter. Inspired by the CLS of human brain, we utilize a hypernetwork to generate adapters to adapt the pre-trained backbone to different tasks. In our framework, the task dictionary functions akin to episodic memory in the hippocampus, while the hypernetwork represents the neocortex, storing past knowledge. The rapidly updating task-specific embeddings and the slowly updating general hypernetwork work together to achieve rehearsal-free continual learning.

of pre-trained models across different tasks. However, it is impractical for learning large numbers of tasks, as the number of adapters involved during inference is proportional to the number of tasks.

In this paper, we propose a novel pre-trained model-based framework named HyperAdapter, for rehearsal-free continual learning. Our method consists of a pre-trained backbone, a hypernetwork, and a set of task embeddings. For any given input, representative features from the pre-trained model are leveraged as queries to identify the most similar task embedding, thus eliminating the necessity of knowing the task identities during inference. Subsequently, the hypernetwork generates a series of adapter parameters based on the obtained task embedding to adapt the pre-trained model to the corresponding task. Our design elegantly addresses the problems of existing methods while inheriting all their advantages. The frozen pre-trained model effectively prevents catastrophic forgetting, adapters enhance the model's expressive ability and adaptive capacity compared to prompts, and the design of hypernetwork avoids the excessive number of adapters in EASE.

According to the Complementary Learning Systems (CLS) theory (Kumaran et al., 2016; McClelland et al., 1995), humans achieve continual learning through the synergy of two systems: the rapidly updating hippocampus focuses on learning task-specific representations, while the slowly updating neocortex specializes in learning more general representations based on past experiences. As shown in Figure 1, task embeddings can be likened to episodic memories in the hippocampus, selectively invoked based on different inputs, while the hypernetwork acts as the neocortex of brain, storing past knowledge in the form of neural connections and updating slowly through the indirect optimization of its generated adapters. Finally, the pre-trained model represents the prior knowledge acquired before learning, aiding the model in better acquiring new tasks. Without a fixed-size prompt pool, the hypernetwork effectively expands model capacity, facilitating positive knowledge transfer. Furthermore, for each new task, our method requires only the addition of a learnable task embedding vector, making it highly suitable for continual learning scenarios involving long sequences and large task numbers. Our empirical results demonstrate that HyperAdapter surpasses the performance of all existing works, offering a significant step forward in rehearsal-free continual learning. In our proposed CL-100 benchmark, HyperAdapter outperformes the previously best DAP and EASE by 2.24% and 3.07% in average final accuracy, respectively. On the larger ImageNet-R and DomainNet, HyperAdapter surpasses the previous SOTA CODA-P and EASE, by 1.06% and 4.80% respectively.

Our main contributions can be summarized as follows:

1. We propose HyperAdapter, a novel rehearsal-free continual learning framework. The method leverages a hypernetwork to generate adapters, adapting the pre-trained model to each task effectively, thereby mitigating forgetting and facilitating positive knowledge transfer.

2. Extensive experiments on various datasets demonstrate that HyperAdapter consistently outperforms all existing methods and even surpasses the multi-task learning upper bound in some cases, establishing a new SOTA for rehearsal-free continual learning. Moreover, the hypernetwork design makes it suitable for continual learning with longer task sequences.

3. To the best of our knowledge, this is the first work leveraging hypernetworks to unlock the potential of pre-trained models in the field of continual learning. We expect that our approach provides a novel perspective on continual learning of pre-trained models.

## 2 RELATED WORK

### 2.1 PRE-TRAINED MODEL-BASED CONTINUAL LEARNING

In recent years, continual learning has emerged as a focal point in the field of machine learning, with the primary challenge being how to incorporate new information effectively without forgetting prior knowledge. Traditional continual learning methods (Rolnick et al., 2019; Rebuffi et al., 2017; Lopez-Paz & Ranzato, 2017; Chaudhry et al., 2018b; Aljundi et al., 2019; Chaudhry et al., 2019) typically rely on the rehearsal of old data, which can raise privacy and storage issues. Although carefully designed regulations (Li & Hoiem, 2017; Lopez-Paz & Ranzato, 2017) have addressed forgetting to some extent, regulation-base methods still underperform rehearsal-based ones.

With advancements in model pre-training, an increasing number of studies begin to explore integrating knowledge from pre-trained models into continual learning, achieving comparable results even without a rehearsal buffer. Some approaches (Wang et al., 2022b;a; Smith et al., 2023; Jung et al., 2023) facilitate continual learning by providing appropriate visual prompts (Jia et al., 2022) to pre-trained models. However, these prompt-based methods typically focus on instance-level enhancements, which offer limited overall benefits to the model. Recently, adapter-based techniques (Zhou et al., 2024; Gao et al., 2024) have shown promising results, outperforming prompt-based ones. The modular design of adapters (Pfeiffer et al., 2021; Chen et al., 2022) allows to retain and leverage the knowledge from pre-trained models more effectively, although they also suffer from limited adaptability and scalability. Leveraging a hypernetwork to generate adapters, HyperAdapter proposes a novel rehearsal free mechanism with the capable of generalization.

### 2.2 COMPLEMENTARY LEARNING SYSTEMS

The Complementary Learning Systems (CLS) theory reveals that humans achieve continual learning through the synergy of two systems that update at different frequencies. Inspired by CLS, several methods (Parisi et al., 2018; Pham et al., 2021; Arani et al., 2022) incorporate multiple networks along with rehearsal buffers, regularization constraints, or other components that expand with the task number. FearNet (Kemker & Kanan, 2017), for instance, employs a three-network structure: one hippocampal network for recalling recent instances, one PFC network for long-term memories, and one additional network for deciding between the two for specific cases. Gomez-Villa et al. (2024) proposes to train an expert network that, unburdened by the task of retaining prior knowledge, focuses on excelling in new tasks. Closer to our approach, Gurbuz et al. (2024) introduces a contextual encoding for rehersal-free scenarios. However, all the methods above do not take pre-trained models into consideration, thus lack of powerful representational capabilities and outstanding performance.

### 2.3 HYPERNETWORKS

Hypernetworks (Ha et al., 2016) were initially proposed for network compression, aimed at generating smaller sets of weights to reduce the computational and storage demands. In efficient fine-tuning (Mahabadi et al., 2021; Zhmoginov et al., 2022; He et al., 2022b; Zhang et al., 2022; Üstün et al., 2022), federated learning (Zhang et al., 2023) and few-shot learning tasks (Sendera et al., 2023), hypernetworks have been employed to dynamically generate parameters, achieving significant performance. These applications demonstrate that hypernetworks can not only adjust models for new tasks but also retain memory of old knowledge. Following this strategy, some hypernet-based continual learning methods (Chandra et al., 2023; Książek & Spurek, 2023; Hemati et al., 2023) generate new parameters for the entire network to adapt to new tasks, but these methods may not fully exploit the prior knowledge of pre-trained models. Different from previous works, HyperAdapter leverages representative features from pre-trained models, thus eliminating necessity of knowing the task identities during inference or the dependence on any rehearsal buffers.

## 3 PRELIMINARIES

### 3.1 CONTINUAL LEARNING

Continual learning requires the model to learn a sequence of tasks in order of arrival. For a sequence of $T$ tasks $\mathcal{D} = \{\mathcal{D}_1, \ldots, \mathcal{D}_T\}$, where each task $\mathcal{D}_t = \{(\boldsymbol{x}, y)\}$ contains tuples of the input sample

$x \in \mathcal{X}$ and its corresponding label $y \in \mathcal{Y}$, a single model $f_{\boldsymbol{\theta}} : \mathcal{X} \to \mathcal{Y}$ parameterized by $\boldsymbol{\theta}$ needs to learn on $\mathcal{D}$ sequentially and predicts the label $y = f(\boldsymbol{x}; \boldsymbol{\theta})$ given input sample $\boldsymbol{x}$ from arbitrary task. Data from previous tasks will not be seen anymore when training future tasks, requiring the model to avoid forgetting old knowledge while learning new tasks. Depending on the differences in task transition, continual learning can be further categorized into multiple settings. In this work, we focus on the more challenging class-incremental learning setting, where task identities are only known during training, which is more common in real-world scenarios.

## 3.2 ADAPTER TUNING

Adapter Tuning, which first emerged in NLP (Houlsby et al., 2019), achieves efficient adaptation to downstream tasks by freezing pre-trained weights and inserting lightweight bottleneck modules into the model. AdaptFormer (Chen et al., 2022) later introduces adapters to visual recognition tasks. The vanilla Vision Transformer consists of a series of blocks, each containing a multi-head self-attention layer (MHSA) and a feed-forward network (FFN). The processing of FFN can be formalized as:

$$\boldsymbol{x} = \text{MLP}(\text{LN}(\boldsymbol{x}')) + \boldsymbol{x}' \tag{1}$$

where $\boldsymbol{x}' \in \mathbb{R}^d$ is the output of MHSA in the same block. AdaptFormer replaces the MLP in FFN with AdaptMLP, which includes an adapter with a down-projection matrix $\boldsymbol{D} \in \mathbb{R}^{d \times r}$ and an up-projection matrix $\boldsymbol{U} \in \mathbb{R}^{r \times d}$, where $r$ is the bottleneck dimension satisfying $r \ll d$. The process of extracting adapted features can be represented as:

$$\tilde{\boldsymbol{x}} = \text{GELU}(\boldsymbol{x}' \cdot \boldsymbol{D}) \cdot \boldsymbol{U} \tag{2}$$

All features are then passed through residual connection to obtain the final output of AdaptMLP:

$$\boldsymbol{x} = \text{MLP}(\text{LN}(\boldsymbol{x}')) + s \cdot \tilde{\boldsymbol{x}} + \boldsymbol{x}' \tag{3}$$

where $s$ is a scale factor to ensure convergence.

## 4 METHOD

We propose HyperAdapter as a hypernetwork-based method for rehearsal-free continual learning. The overall framework is illustrated in Figure 2. We first introduce task-conditional embeddings and the query-key matching mechanism in Section 4.1. Then, we explain how the hypernetwork utilizes these embeddings to generate task-oriented adapters in Section 4.2. And we further extend the hypernetwork to a block-wise implementation in Section 4.3, significantly reducing forgetting and improving the performance. Finally, we present the overall optimization objective in Section 4.4.

## 4.1 TASK-CONDITIONAL EMBEDDINGS

The hypernetwork requires task-conditional embeddings as inputs to generate task-specific adapter parameters. These embeddings serve as episodic memories in the hippocampus, selectively activated based on different inputs, and fed into the hypernetwork storing past knowledge to accomplish any task. To achieve input-dependent selection of task embeddings, we maintain a dictionary $\mathcal{C} = \{(\boldsymbol{k}_1, \boldsymbol{z}_1), (\boldsymbol{k}_2, \boldsymbol{z}_2), \ldots, (\boldsymbol{k}_T, \boldsymbol{z}_T)\}$ of length $T$, where each entry contains a key $\boldsymbol{k} \in \mathbb{R}^d$ and a value $\boldsymbol{z} \in \mathbb{R}^e$, both being learnable parameter vectors. For task embedding selection, we employ a query-key matching mechanism following the previous work. Given a query $\boldsymbol{q} \in \mathbb{R}^d$, we search for the closest key in the dictionary:

$$\tilde{\boldsymbol{k}} = \underset{\boldsymbol{k} \in \boldsymbol{K}}{\arg\min} \, \gamma(\boldsymbol{q}, \boldsymbol{k}) \tag{4}$$

where $\boldsymbol{K} = \{\boldsymbol{k}_i\}_{i=1}^{T}$ is the set of all keys in the dictionary $\mathcal{C}$ and $\gamma$ is the distance metric (we use the negative cosine similarity in our experiments). The corresponding $\tilde{\boldsymbol{z}}$ for the retrieved $\tilde{\boldsymbol{k}}$ is then the task embedding corresponding to that query. This design decouples the learning of keys and values and allows adjusting the total parameters of the hypernetwork through the dimension of task embeddings, providing greater flexibility to the entire framework.

For any given input, we aim for the generated query to be relatively fixed, as this facilitates the learning of task keys. Hence, we simply utilize the pre-trained model as a frozen feature extractor

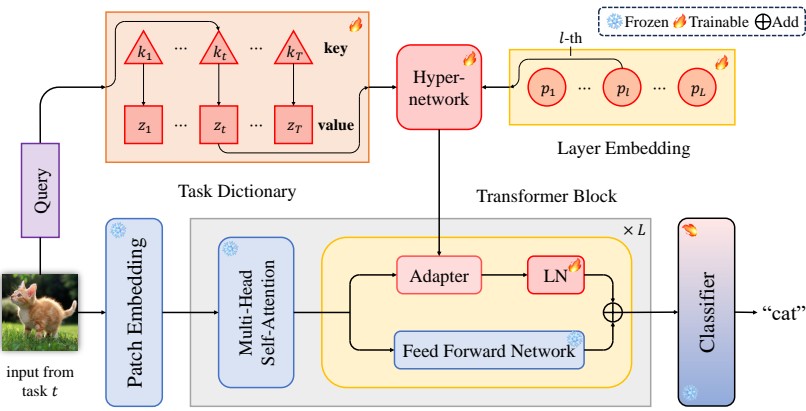

Figure 2: Framework of HyperAdapter. First, inputs from any task are matched to the most similar task embedding through a query. It is then fed into the hypernetwork along with the layer position embedding to generate a set of adapters. The input image is processed through the model with these adapters, and the output is finally obtained via the classifier. We do not update the classifier weights related to previous tasks. And layer embeddings are not included in the block-wise HyperAdapter.

$q(\boldsymbol{x}) = f(\boldsymbol{x})[\text{CLS}]$, where [CLS] represents the class token. In this way, the same input consistently yields the same query, preventing forgetting during the query generation. Moreover, a powerful pre-trained feature extractor also ensures that similar inputs produce similar queries, which aids the model in obtaining the correct task embeddings. And the matching loss can be formulated as:

$$\mathcal{L}_{\text{match}}(\boldsymbol{x}, \boldsymbol{k}_t) = \gamma(q(\boldsymbol{x}), \boldsymbol{k}_t), \quad \boldsymbol{x} \in \mathcal{D}_t \tag{5}$$

During training, since the task identities are known, we use ground truth task embeddings $\boldsymbol{z}_t$ as input to the hypernetwork. In inference, the trained task dictionary $\mathcal{C}$ can be used to obtain input-dependent task embeddings $\tilde{\boldsymbol{z}}$, eliminating the need for task identities and making the entire method applicable to the more challenging class-incremental learning.

## 4.2 TASK-ORIENTED HYPER-ADAPTERS

With the task embeddings, we can adapt the pre-trained model to each task for continual learning using task-oriented hyper-adapters. According to the theory of CLS, humans learn continually through the synergy of two systems. Here we use a hypernetwork to play the role of the neocortex, taking the task embedding $\boldsymbol{z}_t$ as input and generating a set of adapter parameters $\{(\boldsymbol{D}_t^l, \boldsymbol{U}_t^l)\}_{l=1}^L$ for the task $\mathcal{D}_t$. This design enables the hypernetwork to continuously retain past knowledge and promote positive knowledge transfer through information sharing across different tasks.

To increase the model capacity, we introduce a set of positional embeddings $\mathcal{P} = \{\boldsymbol{p}^l \in \mathbb{R}^e\}_{l=1}^L$ for each layer. In practice, we simply add these positional embeddings to the task embedding:

$$\boldsymbol{I}_t^l = \boldsymbol{z}_t + \boldsymbol{p}^l \tag{6}$$

With this combined embedding $\boldsymbol{I}_t^l \in \mathbb{R}^e$, the hypernetwork can generate different parameters for each layer, helping the model better adapt to specific tasks. Furthermore, to prevent forgetting, considering that different $\boldsymbol{p}^l$ are used only to help the hypernetwork distinguish between different layers, we freeze all the positional embeddings $\mathcal{P}$ after learning the first task $\mathcal{D}_1$.

The neocortex stores knowledge in the form of neural connections. Inspired by this, we use a large linear layer as the hypernetwork $h$. Specifically, for the down- and up-projection layers of the adapter, we use two linear layers $\boldsymbol{W}_D \in \mathbb{R}^{(d \times r) \times e}$ and $\boldsymbol{W}_U \in \mathbb{R}^{(r \times d) \times e}$ to generate the adapter parameters:

$$(\boldsymbol{D}_t^l, \boldsymbol{U}_t^l) = h(\boldsymbol{I}_t^l) = (\boldsymbol{W}_D, \boldsymbol{W}_U)\boldsymbol{I}_t^l \tag{7}$$

Additionally, we include an extra layer normalization $\text{LN}^l$ after the adapter to enhance the model's ability to learn new tasks, which is crucial for the performance of continual learning, as detailed in the ablation study. Overall, our task-oriented hyper-adapters can be represented as:

$$\tilde{\boldsymbol{x}} = \text{LN}^l(\text{GELU}(\boldsymbol{x}' \cdot \boldsymbol{D}_t^l) \cdot \boldsymbol{U}_t^l) \tag{8}$$

### 4.3 BLOCK-WISE HYPER-ADAPTERS

While incorporating a single hypernetwork for the entire model is consistent with the original intention of the hypernetwork design, which is to compress the parameters of the network, we find that introducing a separate hypernetwork for each layer significantly enhances the continual learning performance. Since continual learning itself does not impose strict requirements on the number of parameters, this approach effectively decouples task-specific knowledge from position-related knowledge, further mitigating forgetting. Particularly, we introduce a hypernetwork $h^l$ for each layer of the model, comprising two linear layers $\boldsymbol{W}_D^l \in \mathbb{R}^{(d \times r) \times e}$ and $\boldsymbol{W}_U^l \in \mathbb{R}^{(r \times d) \times e}$, to generate adapter parameters for a specific layer:

$$(\boldsymbol{D}_t^l, \boldsymbol{U}_t^l) = h^l(\boldsymbol{z}_t) = (\boldsymbol{W}_D^l, \boldsymbol{W}_U^l)\boldsymbol{z}_t \tag{9}$$

The block-wise hyper-adapters improve performance at the cost of increased parameters. We refer to this method as HA$_{\text{block}}$, whereas the method described in Section 4.2 is referred to as HA$_{\text{model}}$.

### 4.4 OPTIMIZATION OBJECTIVE

During training, for an input $\boldsymbol{x}$ from task $\mathcal{D}_t$, we use the ground truth task key $\boldsymbol{k}_t$ to compute the matching loss in the class-incremental setting. Then we combine the input with the corresponding task embedding $\boldsymbol{z}_t$ and positional embeddings $\mathcal{P}$ (excluded in HA$_{\text{block}}$) and pass it through the pre-trained model $f_{\boldsymbol{\theta}}$ and the hypernetwork $h$. The output is then fed into the classifier $g_{\boldsymbol{\phi}}$ for prediction, are compared with the labels $y$ to calculate the task loss. In summary, the overall optimization objective for HyperAdapter can be expressed as:

$$\min_{\mathcal{C}, \mathcal{P}, \boldsymbol{W}_D, \boldsymbol{W}_U, \boldsymbol{\phi}_t} \mathcal{L}(g_{\boldsymbol{\phi}_t} \circ f(\boldsymbol{x}; \boldsymbol{\theta}, h(\boldsymbol{I}_t; \boldsymbol{W}_D, \boldsymbol{W}_U)), y) + \lambda \mathcal{L}_{\text{match}}(\boldsymbol{x}, \boldsymbol{k}_t), \quad (\boldsymbol{x}, y) \in \mathcal{D}_t \tag{10}$$

where $\mathcal{L}$ represents the task loss, $\mathcal{L}_{\text{match}}$ is the matching loss defined in Section 4.1, and $\lambda$ is a scalar loss weight balancing term (we set it to 0.1 by default). The parameters to be updated include the task dictionary $\mathcal{C}$, the positional embeddings $\mathcal{P}$ (frozen after learning the first task), the hypernetwork $\boldsymbol{W}_D, \boldsymbol{W}_U$, and the classifier $\boldsymbol{\phi}$. Notably, consistent with previous works, we do not update the weights in the classifier related to past tasks, with the remaining parts for the $t$-th task denoted as $\boldsymbol{\phi}_t$.

## 5 EXPERIMENTS

### 5.1 EXPERIMENTAL SETUP

**Datasets.** We conduct extensive experiments on seven datasets with varying data scales and task sequence lengths. The first five datasets are uniformly restricted to 100 classes each and combined to form the CL-100 benchmark. More details of the datasets are provided in the appendix.

- **CL-100 Benchmark**: We select 5 commonly used datasets and combined them into a new benchmark CL-100, for a more comprehensive evaluation. These datasets vary in scale, listed from small to large: Oxford Flowers 102 (Nilsback & Zisserman, 2008), Caltech-101 (Fei-Fei et al., 2006), Stanford Dogs (Dataset, 2011), CIFAR-100 (Krizhevsky et al., 2009), and Food-101 (Bossard et al., 2014). We restrict these datasets to 100 classes each, with each class containing between 20 to 750 training images. For continual learning, the CL-100 benchmark is splitted into 10 tasks by default, each containing 10 classes.

- **Large Benchmarks**: To validate the effectiveness of methods in more challenging scenarios, we include two larger benchmarks: Split ImageNet-R (Hendrycks et al., 2021) and Split DomainNet (Peng et al., 2019). The Split ImageNet-R benchmark is build upon the test set of the ImageNet-R, containing a total of 200 classes and 30,000 images. It is splitted into 10 tasks by default, each with 20 classes. The Split ImageNet-R dataset presents a substantial divergence from the dataset used for backbone pre-training (i.e. ImageNet-21k), and the significant intra-class diversity imposes higher demands on rehearsal-free continual learning methods. The Split DomainNet benchmark comprises over a hundred thousand images across 345 classes from 6 different domains. It is splitted into 15 tasks by default, each containing 23 classes. The Split DomainNet dataset includes images in various styles, which is beneficial for evaluating the generalization ability of model across distinct domains.

**Comparing Baselines.** We compare our method against several rehearsal-free baselines and SOTA methods, including regularization- and architecture-based (e.g. prompt and adapter) approaches. All methods use the same ViT-B/16 backbone pre-trained on ImageNet-21K (Deng et al., 2009). For completeness, we also include two naive baselines. More details can be found in the appendix.

- **Naive Baselines**: We present Full-seq and Linear-seq as two naive baselines. Full-seq denotes the fully sequential training, while Linear-seq refers to a version based on the pre-trained backbone, where only the classifier is updated during sequential training.

- **Regularization-Based Methods**: We choose the classical EWC (Kirkpatrick et al., 2017) and LwF (Li & Hoiem, 2017) as our regularization-based baselines, both of which introduce certain regularization to model parameters during sequential training to prevent forgetting.

- **Prompt-Based Methods**: There are many prompt-based methods utilizing pre-trained models for continual learning. We select L2P (Wang et al., 2022b), DualPrompt (Wang et al., 2022a), CODA-Prompt (Smith et al., 2023), and DAP (Jung et al., 2023) as the current SOTA methods for comparison.

- **Adapter-Based Methods**: We choose EASE (Zhou et al., 2024), the only adapter-based continual learning method prior to our work for a fair comparison.

- **Upper Bound**: Following previous works, we use the multi-task learning results as the upper bound to demonstrate the exceptional performance of our method.

**Evaluation Metrics.** Following DAP, we employ three widely used metrics for method evaluation: final accuracy (Fnl. Acc. ↑) of the accuracy after the last task as final performance, forgetting rate (Forgetting ↓) of the ability to alleviate forgetting (negative transfer inhibition), and learning accuracy (Lrn. Acc. ↑) of the ability to acquire new information (positive transfer promotion). We repeat each experiment three times and report the average values with standard errors. Please refer to the appendix for more details on the evaluation metrics.

**Implementation Details.** We train HyperAdapter using Adam with $\beta_1, \beta_2$ of 0.9, learning rate of 0.01 and batch size of 128. All input images are resized to $224 \times 224$. For smaller datasets, we train every task for 30 epochs to ensure convergence, while for others, we train each task for 10 epochs. In the task dictionary, we maintain only 1 key and 1 embedding for each task. Following previous works, we use a single classifier and do not update the weights corresponding to classes of past tasks. For model-wise HyperAdapter, we set the dimension of both task embeddings and adapter bottleneck to 32, while for block-wise HyperAdapter, we set them to 16. The hyperparameters $s$ in Equation 3 and $\lambda$ in Equation 10 are both set to 0.1. All experiments are conducted on NVIDIA A100 GPUs.

## 5.2 MAIN RESULTS

**CL-100 Benchmark.** Table 1 presents a comprehensive comparison between HyperAdapter and other methods on CL-100 benchmark. Our method consistently outperforms all baselines, establishing a new SOTA for rehearsal-free continual learning. In Table 1(a), which shows the final accuracy, HyperAdapter leds all other methods. Specifically, it surpasses the regularization-based methods EWC and LwF by 37.20% and 32.39% in average accuracy. Moreover, compared to more advanced prompt-based and adapter-based methods, HyperAdapter exceeds the previous best DAP and EASE by 2.24% and 3.07%. The block-wise HyperAdapter further achieves a 1.77% improvement over model-wise HyperAdapter through more parameters. Notably, compared to the multi-task learning upper bound, HyperAdapter even achieved an improvement of 0.78-2.94% on larger datasets (with more than 10,000 training images).

Table 1(b) shows the forgetting performance. With the help of a pre-trained backbone and task dictionary, the model-wise HyperAdapter significantly reduced the forgetting rate by more than 20% compared to regularization-based methods. Benefiting from the decoupling of task and position information, block-wise HyperAdapter further reduced the forgetting rate by 1.11%, reaching an astonishing 1.30%, significantly lower than the previous best DAP (2.26%) and EASE (3.18%). Table 1(c) reflects the ability to learn new tasks based on old knowledge. The design of the hyper-network enables HyperAdapter to better facilitate positive knowledge transfer between tasks. Both versions of HyperAdapter surpass DAP and EASE, as well as Full-seq, in learning accuracy.

Table 1: Results on CL-100 benchmark. Datasets are sorted with their scales in ascending order.

| Dataset | Split Flowers-100 | Split Caltech-100 | Split Dogs-100 | Split CIFAR-100 | Split Food-100 | |
|---|---|---|---|---|---|---|
| **Method** | | (a) Final Accuracy (↑) | | | | **Mean (↑)** |
| Full-seq | $35.64 \pm 1.92$ | $27.04 \pm 1.25$ | $22.67 \pm 0.96$ | $30.39 \pm 1.92$ | $26.90 \pm 0.51$ | 28.53 |
| Linear-seq | $78.95 \pm 0.58$ | $76.17 \pm 0.08$ | $66.22 \pm 0.16$ | $68.43 \pm 0.09$ | $60.58 \pm 0.32$ | 70.07 |
| EWC | $69.79 \pm 1.76$ | $57.96 \pm 1.83$ | $45.72 \pm 1.26$ | $59.60 \pm 1.27$ | $55.27 \pm 1.06$ | 57.67 |
| LwF | $71.78 \pm 1.98$ | $63.26 \pm 1.37$ | $48.97 \pm 1.23$ | $68.22 \pm 1.63$ | $60.15 \pm 0.66$ | 62.48 |
| L2P | $94.53 \pm 1.23$ | $89.34 \pm 1.78$ | $76.59 \pm 1.32$ | $83.05 \pm 1.02$ | $70.48 \pm 1.38$ | 82.80 |
| DualPrompt | $95.25 \pm 0.82$ | $91.52 \pm 0.87$ | $78.36 \pm 0.63$ | $84.77 \pm 0.69$ | $75.31 \pm 0.57$ | 85.04 |
| CODA-P | $97.02 \pm 0.39$ | $91.44 \pm 0.34$ | $84.41 \pm 1.44$ | $86.25 \pm 0.74$ | $77.58 \pm 1.18$ | 87.34 |
| DAP | $96.49 \pm 0.05$ | $97.23 \pm 0.33$ | $87.02 \pm 1.12$ | $94.05 \pm 1.19$ | $88.37 \pm 0.63$ | 92.63 |
| EASE | $97.53 \pm 0.16$ | $96.54 \pm 0.47$ | $88.31 \pm 0.82$ | $87.81 \pm 0.23$ | $88.82 \pm 1.47$ | 91.80 |
| **$HA_{model}$** | $97.57 \pm 0.68$ | $96.44 \pm 0.15$ | $87.14 \pm 0.29$ | $95.85 \pm 0.37$ | $88.64 \pm 0.54$ | 93.13 |
| **$HA_{block}$** | $\mathbf{98.04 \pm 0.42}$ | $\mathbf{97.66 \pm 0.24}$ | $\mathbf{89.32 \pm 0.68}$ | $\mathbf{97.14 \pm 0.35}$ | $\mathbf{92.20 \pm 0.45}$ | **94.87** |
| Upper Bound | $98.83 \pm 0.75$ | $98.32 \pm 0.16$ | $88.54 \pm 0.35$ | $94.20 \pm 0.80$ | $90.40 \pm 1.24$ | 94.06 |
| **Method** | | (b) Forgetting Rate (↓) | | | | **Mean (↓)** |
| Full-seq | $67.53 \pm 0.53$ | $68.97 \pm 1.14$ | $73.12 \pm 1.27$ | $69.21 \pm 0.18$ | $75.70 \pm 0.55$ | 70.91 |
| Linear-seq | $21.93 \pm 0.14$ | $23.66 \pm 0.26$ | $25.78 \pm 0.35$ | $17.34 \pm 0.57$ | $20.69 \pm 0.61$ | 21.88 |
| EWC | $23.73 \pm 3.00$ | $26.34 \pm 3.39$ | $29.26 \pm 1.64$ | $24.65 \pm 0.07$ | $23.67 \pm 1.12$ | 25.53 |
| LwF | $25.14 \pm 2.42$ | $24.63 \pm 0.75$ | $33.42 \pm 1.86$ | $15.44 \pm 1.48$ | $17.15 \pm 0.62$ | 23.16 |
| L2P | $2.12 \pm 0.14$ | $4.86 \pm 0.49$ | $6.93 \pm 0.44$ | $7.21 \pm 0.16$ | $9.09 \pm 0.44$ | 6.04 |
| DualPrompt | $1.40 \pm 0.32$ | $2.85 \pm 0.31$ | $4.84 \pm 0.11$ | $5.60 \pm 0.40$ | $8.76 \pm 0.21$ | 4.69 |
| CODA-P | $1.16 \pm 0.18$ | $2.73 \pm 0.48$ | $4.03 \pm 2.11$ | $4.67 \pm 0.26$ | $7.58 \pm 1.23$ | 4.03 |
| DAP | $0.41 \pm 0.07$ | $0.92 \pm 0.09$ | $2.52 \pm 0.80$ | $2.28 \pm 0.96$ | $5.19 \pm 0.52$ | 2.26 |
| EASE | $0.52 \pm 0.04$ | $1.41 \pm 0.58$ | $3.48 \pm 1.52$ | $5.40 \pm 0.96$ | $5.07 \pm 0.37$ | 3.18 |
| **$HA_{model}$** | $1.25 \pm 0.79$ | $1.45 \pm 0.14$ | $3.56 \pm 0.25$ | $2.08 \pm 0.37$ | $3.73 \pm 1.17$ | 2.41 |
| **$HA_{block}$** | $\mathbf{0.40 \pm 0.12}$ | $\mathbf{0.54 \pm 0.07}$ | $\mathbf{2.08 \pm 0.46}$ | $\mathbf{0.80 \pm 0.44}$ | $\mathbf{2.66 \pm 0.49}$ | **1.30** |
| **Method** | | (c) Learning Accuracy (↑) | | | | **Mean (↑)** |
| Full-seq | $97.93 \pm 1.84$ | $94.87 \pm 2.87$ | $89.48 \pm 0.18$ | $97.75 \pm 1.39$ | $94.04 \pm 0.03$ | 94.81 |
| Linear-seq | $96.01 \pm 0.20$ | $92.58 \pm 0.05$ | $83.20 \pm 1.24$ | $89.50 \pm 1.38$ | $80.20 \pm 0.24$ | 88.12 |
| EWC | $93.47 \pm 1.98$ | $86.29 \pm 0.89$ | $74.23 \pm 1.56$ | $81.78 \pm 1.29$ | $75.49 \pm 0.14$ | 82.25 |
| LwF | $97.14 \pm 0.17$ | $88.01 \pm 0.08$ | $89.51 \pm 0.57$ | $82.05 \pm 0.07$ | $75.59 \pm 0.21$ | 86.46 |
| L2P | $97.37 \pm 0.19$ | $93.54 \pm 0.54$ | $89.78 \pm 0.81$ | $89.13 \pm 0.07$ | $78.65 \pm 0.45$ | 89.69 |
| DualPrompt | $97.72 \pm 0.16$ | $96.52 \pm 0.41$ | $90.11 \pm 0.23$ | $90.61 \pm 0.13$ | $82.82 \pm 0.03$ | 91.56 |
| CODA-P | $98.06 \pm 0.24$ | $96.89 \pm 1.42$ | $90.61 \pm 0.51$ | $91.79 \pm 0.68$ | $87.80 \pm 1.56$ | 93.03 |
| DAP | $96.74 \pm 0.11$ | $97.87 \pm 0.28$ | $89.17 \pm 0.48$ | $96.37 \pm 0.74$ | $93.03 \pm 0.58$ | 94.64 |
| EASE | $98.07 \pm 0.12$ | $96.67 \pm 1.07$ | $90.36 \pm 0.24$ | $92.60 \pm 1.54$ | $93.67 \pm 1.24$ | 94.27 |
| **$HA_{model}$** | $98.24 \pm 0.56$ | $97.70 \pm 0.21$ | $90.26 \pm 0.07$ | $97.71 \pm 0.06$ | $94.83 \pm 0.02$ | 95.75 |
| **$HA_{block}$** | $\mathbf{98.32 \pm 0.49}$ | $\mathbf{98.04 \pm 0.20}$ | $\mathbf{91.10 \pm 0.42}$ | $\mathbf{97.82 \pm 0.07}$ | $\mathbf{94.59 \pm 0.12}$ | **95.97** |

Table 2: Results on Split ImageNet-R and Split DomainNet.

| Dataset | Split ImageNet-R | | | Split DomainNet | | |
|---|---|---|---|---|---|---|
| Method | Fnl. Acc. (↑) | Forgetting (↓) | Lrn. Acc. (↑) | Fnl. Acc. (↑) | Forgetting (↓) | Lrn. Acc. (↑) |
| Full-seq | $21.09 \pm 3.45$ | $54.89 \pm 2.31$ | $75.96 \pm 1.32$ | $27.89 \pm 3.21$ | $72.89 \pm 2.72$ | $89.58 \pm 1.98$ |
| Linear-seq | $55.21 \pm 1.59$ | $19.89 \pm 0.45$ | $74.32 \pm 1.45$ | $72.15 \pm 1.98$ | $12.15 \pm 0.89$ | $84.15 \pm 2.72$ |
| L2P | $60.98 \pm 0.70$ | $9.93 \pm 0.43$ | $69.23 \pm 0.78$ | $80.67 \pm 0.85$ | $5.33 \pm 0.87$ | $85.14 \pm 0.99$ |
| DualPrompt | $68.97 \pm 2.87$ | $4.66 \pm 2.15$ | $72.85 \pm 2.27$ | $81.89 \pm 0.63$ | $5.21 \pm 1.17$ | $87.27 \pm 1.80$ |
| CODA-P | $75.45 \pm 0.56$ | $\mathbf{1.64 \pm 0.10}$ | $77.90 \pm 1.97$ | $76.52 \pm 0.78$ | $6.83 \pm 0.10$ | $82.53 \pm 0.24$ |
| DAP | $70.12 \pm 2.24$ | $2.90 \pm 2.70$ | $73.24 \pm 2.81$ | $83.51 \pm 1.07$ | $5.30 \pm 0.52$ | $88.77 \pm 0.79$ |
| EASE | $73.23 \pm 1.65$ | $6.79 \pm 1.19$ | $76.97 \pm 1.25$ | $86.76 \pm 1.29$ | $4.67 \pm 0.92$ | $91.74 \pm 1.28$ |
| **$HA_{model}$** | $75.65 \pm 2.67$ | $5.78 \pm 2.94$ | $\mathbf{79.22 \pm 0.96}$ | $89.20 \pm 0.54$ | $4.18 \pm 0.49$ | $93.10 \pm 0.24$ |
| **$HA_{block}$** | $\mathbf{76.51 \pm 1.32}$ | $3.83 \pm 1.46$ | $77.97 \pm 0.81$ | $\mathbf{91.56 \pm 0.11}$ | $\mathbf{2.18 \pm 0.10}$ | $\mathbf{93.58 \pm 0.12}$ |
| Upper Bound | $77.13 \pm 1.54$ | - | - | $90.65 \pm 0.98$ | - | - |

Interestingly, the sequential learning method that only updates the classifier (known as linear probing) also reaches an impressive final accuracy of 70.07%, surpassing the classic regularization-based methods EWC and LwF, which fully demonstrates the tremendous potential of pre-trained models.

**ImageNet-R and DomainNet.** To further demonstrate the performance of HyperAdapter, we conduct experiments on two larger benchmarks: Split ImageNet-R and Split DomainNet, with results shown in Table 2. On both datasets, HyperAdapter surpasses the previous best methods, CODA-P and EASE, by 1.06% and 4.80%, respectively. This fully demonstrates the capability of HyperAdapter to handle more categories and longer task sequences. Moreover, HyperAdapter closely approaches the upper bound of multi-task learning on ImageNet-R and even surpasses the upper bound on the larger DomainNet dataset. Experimental results on longer sequences are provided in the appendix.

Table 3: LoRA results on CL-100 benchmark. HLA stands for the LoRA version of HyperAdaper.

| Dataset | Split Flowers-100 | Split Caltech-100 | Split Dogs-100 | Split CIFAR-100 | Split Food-100 | |
|---|---|---|---|---|---|---|
| Method | (a) Final Accuracy (↑) | | | | | Mean (↑) |
| $\text{HLA}_{\text{model}}$ | $95.86 \pm 0.00$ | $93.13 \pm 0.49$ | $86.07 \pm 0.08$ | $92.20 \pm 0.69$ | $87.64 \pm 0.18$ | 90.98 |
| $\text{HLA}_{\text{block}}$ | $96.21 \pm 1.21$ | $97.93 \pm 0.14$ | $90.04 \pm 0.13$ | $97.80 \pm 0.01$ | $93.72 \pm 0.14$ | 95.14 |
| Method | (b) Forgetting Rate (↓) | | | | | Mean (↓) |
| $\text{HLA}_{\text{model}}$ | $2.24 \pm 0.03$ | $0.45 \pm 1.08$ | $5.64 \pm 0.28$ | $2.06 \pm 0.93$ | $3.73 \pm 0.21$ | 2.82 |
| $\text{HLA}_{\text{block}}$ | $1.18 \pm 0.33$ | $0.54 \pm 0.03$ | $2.06 \pm 0.07$ | $0.40 \pm 0.00$ | $2.14 \pm 0.14$ | 1.30 |
| Method | (c) Learning Accuracy (↑) | | | | | Mean (↑) |
| $\text{HLA}_{\text{model}}$ | $99.22 \pm 0.01$ | $97.85 \pm 0.08$ | $91.03 \pm 0.10$ | $97.60 \pm 0.00$ | $94.11 \pm 0.03$ | 95.96 |
| $\text{HLA}_{\text{block}}$ | $99.32 \pm 0.38$ | $98.12 \pm 0.06$ | $91.72 \pm 0.00$ | $98.18 \pm 0.00$ | $95.65 \pm 0.02$ | 96.60 |

Table 4: Ablation studies on core designs.

| Dataset | Split CIFAR-100 | | | Split ImageNet-R | | |
|---|---|---|---|---|---|---|
| Method | Fnl. Acc. (↑) | Forgetting (↓) | Lrn. Acc. (↑) | Fnl. Acc. (↑) | Forgetting (↓) | Lrn. Acc. (↑) |
| $\text{HA}_{\text{model}}$ | $95.85 \pm 0.37$ | $2.08 \pm 0.37$ | $97.71 \pm 0.06$ | $75.65 \pm 2.67$ | $5.78 \pm 2.94$ | $79.22 \pm 0.96$ |
| w/o task dictionary | $66.32 \pm 1.61$ | $33.46 \pm 1.84$ | $96.43 \pm 0.06$ | $45.48 \pm 5.35$ | $35.06 \pm 5.35$ | $77.19 \pm 5.58$ |
| w/o position embedding | $94.43 \pm 0.11$ | $2.30 \pm 0.17$ | $96.57 \pm 0.11$ | $69.18 \pm 1.20$ | $9.09 \pm 0.74$ | $76.39 \pm 2.29$ |
| w/o adapter LN | $29.58 \pm 2.73$ | $22.84 \pm 1.49$ | $50.13 \pm 3.89$ | $13.01 \pm 0.46$ | $6.80 \pm 0.88$ | $18.85 \pm 0.69$ |

## 5.3 HYPERADAPTER WITH LORA

To verify that the hypernetwork-based design of our HyperAdapter is not confined to any specific adapter structure, we choose Low-Rank Adaptation or LoRA (Hu et al., 2021), an adapter structure initially proposed for efficiently fine-tuning large language models. We refer to the LoRA version of our approach as HyperLoRA or HLA, which is also divided into model-wise and block-wise versions. The results on CL-100 benchmark are shown in Table 3. In this experiment, we followed the original settings in Hu et al. (2021), applying LoRA to the query and value matrices in ViT's attention mechanism, with the rank set to half of the bottleneck dimension in HA's adapter to ensure that the learnable parameters of there two versions are consistent. All other hyperparameters remained the same as in the HA experiments. The results show that our method also achieves excellent performance when applied to LoRA, even outperforming the HA version without hyperparameter tuning, fully demonstrating the versatility of our design.

## 5.4 ABLATION STUDIES

**Core Design.** To validate the effectiveness of the core designs in HyperAdapter, we conduct ablation studies on the Split CIFAR-100 and ImageNet-R datasets, all based on the model-wise version. As shown in Table 4, row 1 represents the complete version of HyperAdapter. In row 2, we remove the task dictionary and use an additional MLP to project queries to the specified dimension, providing instance embeddings to the hypernetwork as input. This MLP is frozen after learning the first task. The results indicate that removing task embeddings has a minor impact on learning accuracy, but overly diverse inputs easily lead to forgetting in the hypernetwork, resulting in $\sim 30\%$ drop in final accuracy. Row 3 removes the position embedding from each layer, causing the hypernetwork to always generate the same adapter. While repeated adapters have a minimal effect on model performance, due to the minimal parameter count of position embeddings, we ultimately chose the performance improvement brought by diverse adapters. In row 4, we remove the layer normalization (LN) after each adapter. The results demonstrate that this design has a significant impact on the final performance, with removing it resulting in over 60% performance loss. Since learning accuracy also decreases, it indicates that the LN is more for stabilizing the training of the adapters.

**Parameter Scale.** In Figure 3, we show the impact of parameter scale on performance. Figure 3(a) and (b) represent two different versions of HyperAdapter. It can be observed that model performance is positively correlated with parameter scale, which means more parameters lead to higher accuracy. However, as the parameter scale continues to increase, this marginal benefit gradually diminishes. To strike a balance between parameters and performance, we set both the task embedding and adapter bottleneck dimension of model-wise HyperAdapter to 32, while both dimensions of block-wise HyperAdapter are set to 16. Figure 3(c) demonstrates the performance change when fixing one

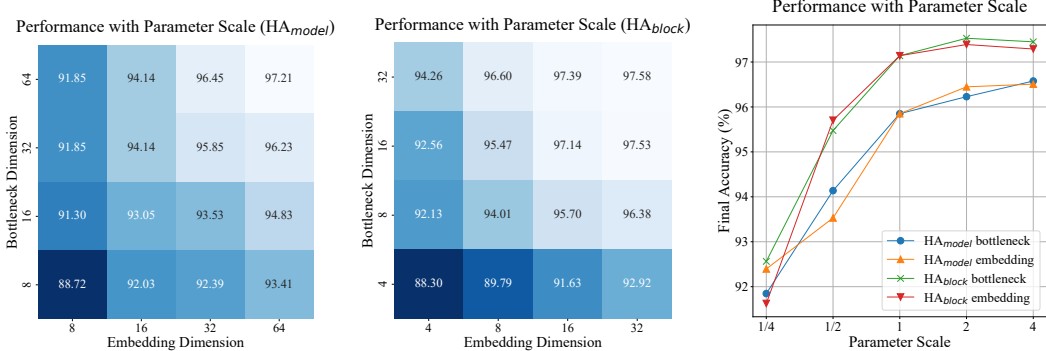

Figure 3: Further analysis on parameter scales on Split CIFAR-100.

dimension and only altering the other one. It can be seen that when the parameter scale reaches a certain threshold, the final performance no longer increases and may even decrease.

## 5.5 VISUALIZATIONS

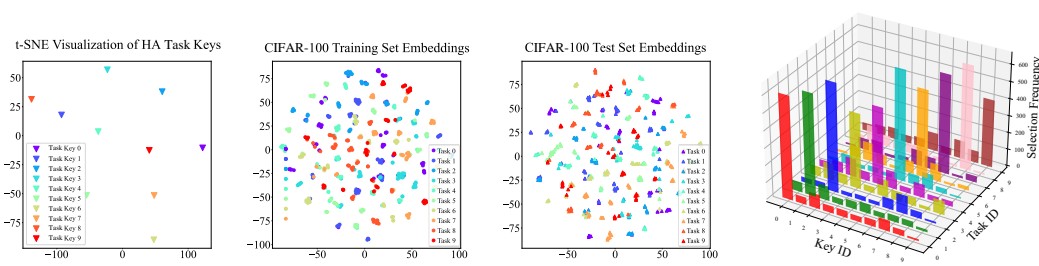

Figure 4: Visualizations on task embeddings and the selection on Split CIFAR-100.

**t-SNE.**  In Figure 4(a), we present t-SNE results of input queries and task keys. To enable the hypernetwork to generate corresponding adapters for different tasks, we adopt a query-key matching mechanism to obtain task embeddings, which can be viewed as clustering based on the pre-trained model. Due to the gap between pre-training data and downstream tasks, the pre-trained model may not always distinguish inputs from different classes well. Moreover, since a task comprises data from different classes, the large inter-class gap can make the clustering process more challenging. Nevertheless, this matching mechanism still plays a crucial role in our method, as shown in Table 4.

**Task Embedding Selection.**  In Figure 4(b), we further show the instance-level selection frequency of different task embeddings. Most tasks correctly select their corresponding embeddings, and the noise from a few incorrect selections do not significantly impact model performance, benefiting from the similarity-based matching mechanism. Even if the matching is incorrect, the matched task is still highly similar, and the embedding can provide enough information for the current task to achieve correct classification. Improving this selection mechanism is left as a direction for future work.

## 6 CONCLUSION

In this paper, we propose HyperAdapter as a novel rehearsal-free continual learning method, utilizing a hypernetwork to generate adapters to adapt the pre-trained model to different tasks. This work represents the first attempt to employ a hypernetwork for continual learning based on pre-trained models, providing a feasible approach for the continual learning of large-scale pre-trained models. We establish a new SOTA on our newly curated CL-100 benchmark and two commonly used large benchmarks, even surpassing the multi-task learning upper bound in some cases. This highlights the immense potential of pre-trained models in continual learning and marks a significant step forward in the application of neural networks in real-world scenarios.

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
