# APPENDIX

In this appendix, we present detailed information about the datasets, comparing baselines and evaluation metrics, along with additional experimental results of HyperAdapter on longer task sequences as well as parameter and computation efficiency comparison with other methods. In Section A, we show details about all the CL benchmark datasets. In Section B, we provide details about comparing baselines and other SOTA method. In Section C, we provide details about evaluation metrics used in experiments. In Section D, we conduct additional experiments of HyperAdapter on Split DomainNet with longer task sequences. In Section E and F, we show the parameter and computation efficiency comparison between our HyperAdapter and other comparable methods.

## A  ADDITIONAL DETAILS OF DATASET

In this paper, we utilize seven datasets with varying levels of dataset scale. Table 5 summarizes the used datasets, number of classes, number of tasks, number of training and test images. Furthermore, we introduce two benchmarks with a longer task sequence that includes a relatively large number of classes to validate the superior performance of HyperAdapter on larger or/and longer benchmarks.

Table 5: Specifications of the various CL benchmarks evaluated.

| Dataset | # Classes | # Tasks | Train | Validation | Test |
|---|---|---|---|---|---|
| Split Flowers-100 | 100 | 10 | 1600 | 400 | 6083 |
| Split Caltech-100 | 100 | 10 | 2400 | 600 | 5617 |
| Split Dogs-100 | 100 | 10 | 8000 | 2000 | 7028 |
| Split CIFAR-100 | 100 | 10 | 40000 | 10000 | 10000 |
| Split Food-100 | 100 | 10 | 50000 | 25000 | 25000 |
| Split ImageNet-R | 200 | 10 | 18000 | 6000 | 6000 |
| Split DomainNet | 345 | 15 | 96724 | 24182 | 52041 |

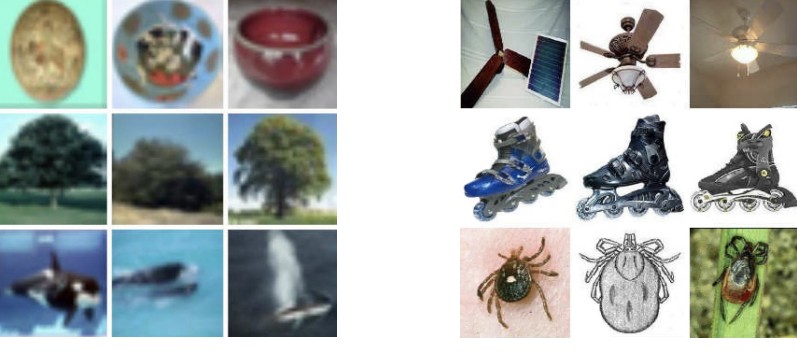

CIFAR-100                                    Caltech-101

Figure 5: Samples of CIFAR-100 and Caltech-101. Each row shows samples from the same class.

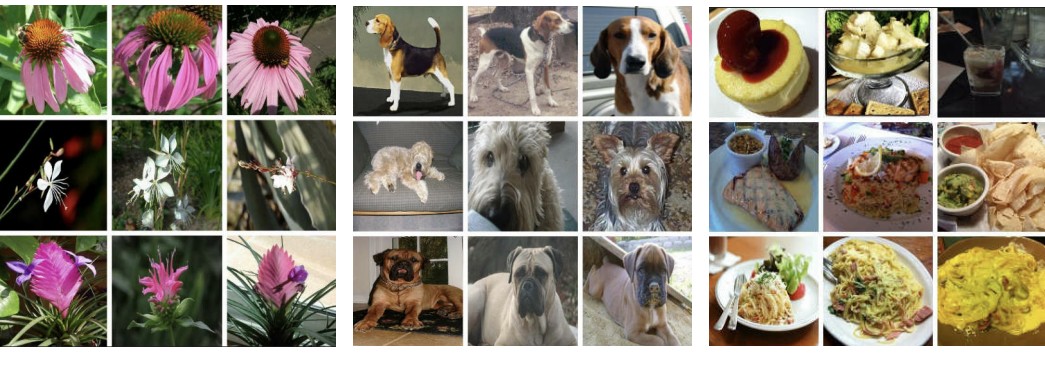

Flowers                          Dogs                          Food-101

Figure 6: Samples of Flowers, Dogs and Food-101. Each row shows samples from the same class.

Here, a description of each benchmark is provided below:

- **CL-100 Benchmark:** To help better understand CL-100 Benchmark, we provide representative examples of all CL-Benchmarks at Figure 5 and Figure 6.

    1. The original Flowers-102 dataset (Nilsback & Zisserman, 2008) contains 102 flower categories with a total of 8189 images. To create Split Flowers-100, we exclude 2 categories, resulting in a dataset with 100 categories, and divide it into a training set of 2000 images and a test set of 6083 images.
    2. The original Caltech-101 dataset (Fei-Fei et al., 2006) consists of 101 categories and 9146 images. We adjust this dataset to create Split Caltech-100 by removing 2 categories, leading to 100 categories with a training set of 3000 images and a test set of 5617 images.
    3. The original Dogs-100 (Dataset, 2011) dataset contains 100 dog categories with 12000 training images and 8580 test images. We modify this dataset to maintain the same 100 categories but with 10000 training images and 7028 test images.
    4. The CIFAR-100 (Krizhevsky et al., 2009) dataset originally includes 100 categories with 50000 training images and 10000 test images. We keep the structure intact for Split CIFAR-100, maintaining 100 categories with the same number of training and test images.
    5. The original Food-101 dataset (Bossard et al., 2014) comprises 101 food categories with a total of 75750 training images and 25250 test images. For Split Food-100, we exclude 1 category, resulting in 100 categories, and create a training set of 75000 images and a test set of 25000 images.

- **Large Benchmark:** We incorporate two benchmarks with a substantial number of classes to showcase the robustness of HyperAdapter in managing large-scale datasets.

    1. ImageNet-R (Hendrycks et al., 2021) is a collection encompassing 200 classes from ImageNet, featuring various artistic renditions such as graffiti, origami, paintings, and sketches. As shown in Table 5, this benchmark originates from the 200 original ImageNet classes used for pre-training the ViT model. Due to this, its domain similarity to ImageNet remains high. The primary objective of including this benchmark is to evaluate scalability concerning dataset size rather than domain adaptation. Split ImageNet-R is constructed by partitioning the 200 classes into 10 distinct tasks, each comprising 20 unique classes.
    2. DomainNet (Peng et al., 2019) consists of images from six different types, totaling 345 categories. For our experiments, we focus on real-type images to create the Split DomainNet benchmark. This benchmark is employed to test the model's robustness over a large number of classes and extended sequences. Split DomainNet is utilized in two configurations: one where the 345 classes are divided into 15 tasks, each containing 23 distinct classes, and another where they are divided into 69 tasks, each containing 5 distinct classes.

## B  ADDITIONAL DETAILS OF COMPARING BASELINES

To verify the relative effectiveness of all methods, we include FT-seq, the naive sequential training approach (considered the lower bound), and the upper bound, which represents supervised joint fine-tuning on the combined data of all tasks. In order to emphasize the ability of PTMs to help continuous learning, we add FT-Linear baseline and only fine-tuned the head of the model pre-trained on Imagenet to ensure a fair comparison.

EWC (Kirkpatrick et al., 2017), a prominent algorithm in continual learning, addresses catastrophic forgetting by regularizing the model's weights based on Fisher information. For fairness, we initialize the model weights from an ImageNet pre-trained model. Similarly, LwF (Li & Hoiem, 2017) employs distillation loss to mitigate catastrophic forgetting, and is a well-established baseline in continual learning. Here we start with ImageNet pre-trained weights for a fair comparison. L2P (Wang et al., 2022b) is the pioneering prompt-based method in continual learning. It utilizes a shared prompt pool to adapt to incoming sequential tasks using a pre-trained model. For consistency and fair comparison,

we employ the same pre-trained model in our method. In contrast, DualPrompt (Wang et al., 2022a) introduces a different prompt-based approach. It distinguishes itself from L2P by employing two types of prompts with distinct objectives: task-invariant and task-agnostic. This method leverages both types of prompts to enhance adaptability across various tasks. CODA-P (Smith et al., 2023) proposes to learn a set of prompt components which are assembled with input-conditioned weights to produce input-conditioned prompts, resulting in a novel attention-based end-to-end key-query scheme. DAP (Jung et al., 2023), a pool-free approach that generates a suitable prompt in an instance-level manner at inference time. Currently, it is the state-of-the-art prompt-based method in continual learning. EASE (Zhou et al., 2024), serving as the only adapter-based baseline accepted by CVPR 2024, train a distinct lightweight adapter module for each new task, aiming to create task-specific subspaces. We use the same pre-trained model for a fair comparison.

## C   ADDITIONAL DETAILS OF EVALUATION METRICS

1. **Average Accuracy:** As outlined in Chaudhry et al. (2018a), average accuracy is defined as the mean accuracy over all tasks after the model has been trained on the final task $T$. It is a widely adopted metric in continual learning, and the metric can be formulated as:

$$\text{Avg Acc} = a_T \quad \text{where} \quad a_i = \frac{1}{i} \sum_{j=1}^{i} a_{i,j},$$

where $a_{i,j}$ represents the accuracy on the test set of the $j$-th task when the model is trained up to the $i$-th task.

2. **Forgetting Measure:** The forgetting measure (Chaudhry et al., 2018a) quantifies the difference between the maximum performance on previous tasks and the performance on those tasks after subsequent training. It estimates how much the model forgets prior tasks $j$ when training on a new task $k$ (with $k > j$). It can be defined as:

$$\text{Forgetting} = \frac{1}{T-1} \sum_{j=1}^{T-1} f_j^T \quad \text{where} \quad f_j^k = \max_{l \in \{1,2,\ldots,k-1\}} a_{l,j} - a_{k,j}.$$

3. **Learning accuracy:** Referenced in Riemer et al. (2019), learning accuracy measures the model's ability to acquire new knowledge from incoming tasks. It is calculated as the mean accuracy of each task immediately after training on it, expressed as:

$$\text{Lrn Acc} = \frac{1}{T} \sum_{j=1}^{T} a_{j,j}.$$

## D   LONGER TASK SEQUENCES RESULTS

Table 6: Results on Split DomainNet with 15/69 tasks.

| Benchmark | DAP | | | HA$_{model}$ | | | HA$_{block}$ | | |
|---|---|---|---|---|---|---|---|---|---|
| | Avg Acc (↑) | Forgetting (↓) | Lrn Acc (↑) | Avg Acc (↑) | Forgetting (↓) | Lrn Acc (↑) | Avg Acc (↑) | Forgetting (↓) | Lrn Acc (↑) |
| 15-Split DomainNet | $83.51 \pm 1.07$ | $5.30 \pm 0.52$ | $88.77 \pm 0.79$ | $89.20 \pm 0.54$ | $4.18 \pm 0.49$ | $93.10 \pm 0.24$ | $\mathbf{91.56 \pm 0.11}$ | $\mathbf{2.18 \pm 0.10}$ | $\mathbf{93.58 \pm 0.12}$ |
| 69-Split DomainNet | $83.36 \pm 0.81$ | $6.75 \pm 1.72$ | $90.50 \pm 0.79$ | $87.16 \pm 0.39$ | $6.95 \pm 0.31$ | $93.80 \pm 0.10$ | $\mathbf{90.05 \pm 0.09}$ | $\mathbf{4.80 \pm 0.19}$ | $\mathbf{94.58 \pm 0.21}$ |

To validate the performance of our method in continual learning with longer task sequences, we also conducted experiments on the 69-Split DomainNet dataset, with results shown in Table 6. Even with such a large number of tasks, our HyperAdapter consistently achieved optimal performance (90.05%), significantly outperforming DAP (83.36%). Furthermore, compared to experiments with a 15-task partition (91.56%), there was no noticeable decline in performance, further demonstrating that our design is well-suited for continual learning with long task sequences.

## E   PARAMETER EFFICIENCY COMPARISON

The parameter efficiency comparison results of different methods are shown in Table 7. From this table, we can draw the following observations:

Table 7: **Parameter efficiency comparison.** Mean Acc. denotes the mean final accuracy on the Continual-100 benchmark. Learnable Params. indicates the total number of learnable parameters. Percentile Params. represents the proportion of learnable parameters relative to the total parameters of the pre-trained backbone. Relation outlines the connections between the learnable parameters and various hyperparameters. Hyperparameters display the specific values of the hyperparameters involved in each method, where $\Theta$ denotes the backbone, $d$ is the backbone embedding size (768), $e$ is the task embedding dimension, $k$ represents the ratio of pool size to the task number, $n$ is the token number, $p$, $p_g$, and $p_e$ represent the lengths of the normal, general, and expert prompts, respectively, $r$ is the bottleneck dimension in the adapter, $C$ is the class number, $L$, $L_g$, and $L_e$ are the layers applied in each method, and $T$ denotes the task number. In the line of EASE, the parentheses indicate parts that are not learnable but occupy memory.

| Method | Mean Acc. (%) | Learnable Params. (M) | Percentile Params. (%) | Relation | Hyperparameters |
|---|---|---|---|---|---|
| Full-seq | 28.53 | 85.80 | 100.00 | $\Theta$ | - |
| Linear-seq | 70.07 | 0.00 | 0.00 | $0$ | - |
| EWC | 57.67 | 85.80 | 100.00 | $\Theta$ | - |
| LwF | 62.48 | 85.80 | 100.00 | $\Theta$ | - |
| L2P | 82.80 | 0.05 | 0.05 | $dk(p+1)T$ | $k=1, p=5$ |
| DualPrompt | 85.04 | 0.48 | 0.55 | $dp_g L_g + d(p_e L_e + 1)T$ | $p_g = 5, L_g = 2, p_e = 20, L_e = 3$ |
| CODA-P | 87.34 | 3.23 | 3.76 | $dk(pL+2)T$ | $k=10, p=8, L=5$ |
| DAP | 92.63 | 0.36 | 0.42 | $((n+1)p+2d(e+2))L + (d+e)T$ | $e=16, n=196, p=10, L=12$ |
| EASE | 91.80 | 2.95(+7.68) | 3.44(+8.95) | $2drLT + dCT^2$ | $r=16, L=12$ |
| HA$_{\text{model}}$ | 91.13 | 0.42 | 0.49 | $2der + (2d+e)L + (d+e)T$ | $e=16, r=16, L=12$ |
| HA$_{\text{model}}$ | 93.13 | 1.60 | 1.86 | $2der + (2d+e)L + (d+e)T$ | $e=32, r=32, L=12$ |
| HA$_{\text{block}}$ | 93.72 | 4.74 | 5.53 | $2d(er+1)L + (d+e)T$ | $e=16, r=16, L=12$ |
| Upper Bound | 94.06 | 85.80 | 100.00 | $\Theta$ | - |

1. Pre-trained model-based methods have a relatively small number of learnable parameters. Notably, HyperAdapter can achieve competitive performance with only 0.5% of the parameters. With just 1.9% of parameters, HyperAdapter significantly outperforms all other methods, and with 5.5% parameters, it even surpasses the multi-task learning upper bound.

2. In experiments on model scalability from other works, increasing the number of learnable parameters does not significantly improve performance. However, in our method, this increase markedly enhances the model's performance, and this trend shows no signs of saturation, indicating that the hypernetwork-based approach has great potential for scalability.

3. Among all methods, the number of learnable parameters in our HyperAdapter shows the smallest variation with the number of tasks. This means that for a new task, HyperAdapter can be accomplished with minimal cost, which is highly valuable for deploying models in real-world scenarios with thousands of complex tasks.

# F   COMPUTATIONAL EFFICIENCY COMPARISON

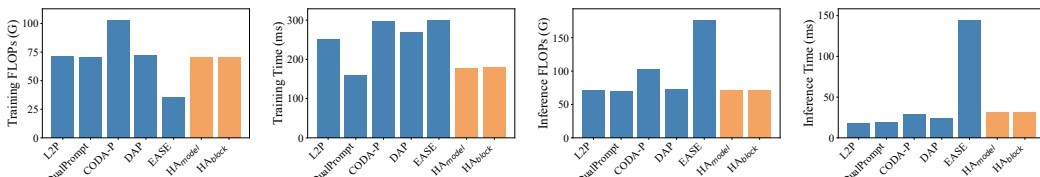

Figure 7: **Computational efficiency comparison.** From left to right: training FLOPs, training time, inference FLOPs, and inference time. FLOPs are calculated on instance-level input. Time costs represent the average cost of processing a batch with size 32, measured on a single A100-80GB GPU.

We have analyzed the runtime costs of different methods in Figure 7. Overall, HyperAdapter maintains similar FLOPs and time costs to other existing methods while achieving significantly better performance. During the training phase, only EASE does not use the query-key matching mechanism, resulting in the lowest FLOPs. Other methods include two forward passes of the backbone, making their FLOPs approximately twice that of EASE. CODA-P introduces an attention mechanism in the prompt, which adds extra computation, resulting in higher FLOPs than other methods. The time cost also considers the parameter update process, with DualPrompt and HyperAdapter taking less time but showing no significant difference. During the inference phase, EASE's required forward passes are related to the number of tasks, resulting in the highest FLOPs. The inference time costs follow a similar trend to FLOPs, with HyperAdapter being comparable to other prompt-based methods.