# OpenReview forum: "HyperAdapter: Generating Adapters for Pre-Trained Model-Based Continual Learning"
_ICLR.cc/2025/Conference — ICLR 2025 Conference Withdrawn Submission_

### Official Review · Reviewer_4wJE · 2024-10-17

**Soundness:** 2
**Presentation:** 3
**Contribution:** 3
**Rating:** 5
**Confidence:** 5

**Summary:**

This paper proposed pre-trained model-based continual learning method that employs a hypernetwork to generate adapters based on the current input. The proposed method features positive transfer and fewer additional parameters as the number of tasks increases. It outperforms a variety of continual learning methods in many representative benchmarks.

**Strengths:**

1. This paper is essentially well-organized and easy to follow.

2. The proposed hypernetwork seems to be a simple but effective strategy, applicable to both adapter-based and LoRA-based parameter-efficient tuning.

3. Less requirement for additional parameters is a good feature for continual learning that ensures scalability.

**Weaknesses:**

1. As acknowledged by the authors, the hypernetwork itself is a large linear layer, and the use of a separate hyper parameter for each layer results in much more parameter cost. For fairness, it is therefore desirable to compare the total parameter cost with other baselines methods.

2. Does the hypernetwork need to be updated in continual learning? If so, how does it overcome catastrophic forgetting?

3. The authors only considered one particular pre-trained checkpoint of supervised ImageNet-21K. Does the proposed method apply to other pre-trained checkpoints, especially for self-supervised pre-training?

4. The authors compared only a representative selection of pre-trained model-based continual learning methods. It would be more informative to consider other concurrent competitors, such as SLCA (ICCV’23), LAE (ICCV’23), RanPAC (NeurIPS’23), HiDe (NeurIPS’23), etc.

**Questions:**

Please refer to the Weaknesses.

---

### Official Review · Reviewer_bCqR · 2024-11-02

**Soundness:** 3
**Presentation:** 1
**Contribution:** 2
**Rating:** 3
**Confidence:** 4

**Summary:**

The paper addresses the problem of catastrophic forgetting in continual learning. The authors introduce a pre-trained model-based continual learning framework, HyperAdapter, which utilizes a hypernetwork to generate adapters based on the current input, adapting the pre-trained model to the corresponding task. A key to the method is that HyperAdapter uses representative features from pre-trained models, eliminating the necessity to know the task identities during inference or the dependence on any rehearsal buffers. Experimentation shows that it outperforms other methods.

**Strengths:**

The research will be of interest to the ICLR community.

Originality: There is currently a lot of interest in avoiding catastrophic forgetting in the continual learning setting. The authors have summarised and categorised the main approaches.

Experimentation: The experimentation is carried out on the standard data sets.

Reproducibility: I believe that the details are sufficient for reproducibility of the experiments.

**Weaknesses:**

Clarity: The paper discusses the main approaches at a high level and fails to clearly describe the key novelty of the proposed approach. Additionally, the comparison of the proposed method with how people learn is repeated in a number of places. The points around this are well known and widely documented. Removing the replication provides space to describe the novelty in more detail and room to discuss the implications of the results.

Typos: Please check your paper carefully for typos including the repeated use of “regularation” on page 3. A grammar checker will pick up some of the errors.

Discussion of results: The paper is missing a discussion of the results. Adding this will provide a deeper understanding of the advantages of the approach.

Discussion of limitations:  The paper is missing a discussion of the limitations of the approach and potential ways to address them. Adding this will provide a more balanced presentation of the research.

Discussion of braider impact: What are the open problems or future directions related to your work? Adding this to the paper would improve the paper's discussion of broader impact and potential future work.

**Questions:**

The values of the hyperparameters are stated on page 7. How does varying the hyperparameter values affect performance?

What are the memory requirements, and what do they depend on?

What are the time requirements, and what do they depend on?

“Improving this selection mechanism is left as a direction for future work.” – can you suggest some possible directions for improvement?

---

### Official Review · Reviewer_j536 · 2024-11-03

**Soundness:** 3
**Presentation:** 3
**Contribution:** 2
**Rating:** 5
**Confidence:** 4

**Summary:**

This paper proposes a class-incremental continual learning framework HyperAdapter, which uses hypernetworks to generate adapters to adapt a pre-trained model to different tasks. Specifically, it uses the pre-trained model's class embedding as a key to query the task embedding that generates the adapter parameters. Extensive experiments are performed on image classification benchmarks, showing improved performance over other regularization-based and prompt-based CL frameworks.

**Strengths:**

- Generates parameters specifically for pre-trained model adapters rather than the entire network, enhancing training efficiency.
- Introduces a task embedding dictionary for efficient retrieval of task embeddings for incoming tasks.
- Provides a thorough and detailed experimental analysis.

**Weaknesses:**

- The approach appears to be a direct adaptation of a hypernet-based continual learning framework to adapter-based fine-tuning. Task relationships rely solely on the pre-trained model’s class token embedding for input, making it resemble a method of training separate models for distinct classes, with conditional parameter generation handled by the hypernet. This setup may not effectively handle scenarios where new classes can not be easily matched with labels of the pre-trained model, such as domain-specific tasks.  e.g. for complex facial emotion classification tasks, the pre-trained model would give similar class embeddings (e.g. human, face, eye glasses, etc) regardless of which emotion class the image belongs to.

- The paper draws an analogy between the proposed method and the brain’s Complementary learning system (CLS). However, unlike the human brain, which can dynamically adjust its internal representations, such as merging or acquiring new concepts, the task dictionary here has keys fixed by the pre-trained classes, lacking the true flexibility of dictionary learning to adapt and integrate new concepts. It's suggested to consider ways to make the task dictionary more dynamic or adaptive over time.

**Questions:**

- How is the task dictionary initialized? The definition of $q(x)$ in Section 4.1 requires more clarity: is [CLS] a one-hot vector, or does it represent the class token embedding of ViT (which is then multiplied by f(x))?

- Does the size of the task dictionary correspond to the number of training tasks (i.e., classification tasks) or the total number of classes across these tasks?

- How is Equation 10 optimized? Including the training algorithm or detailed description of the training process of different model parts would be beneficial.

- What is the parameter scale unit in Figure 3? Does it measure the parameters of the hypernetwork or the generated model parameters (e.g.,  U,W) during inference?

- How does the proposed method compare with other hypernetwork-based continual learning approaches?
e.g.
    * Ding, F., Xu, C., Liu, H., Zhou, B., & Zhou, H. (2024). Bridging pre-trained models to continual learning: A hypernetwork based framework with parameter-efficient fine-tuning techniques. Information Sciences, 674, 120710.
    * Hemati, Hamed, Vincenzo Lomonaco, Davide Bacciu, and Damian Borth. "Partial hypernetworks for continual learning." In Conference on Lifelong Learning Agents, pp. 318-336. PMLR, 2023.

---

### Official Review · Reviewer_c4Eb · 2024-11-04

**Soundness:** 2
**Presentation:** 3
**Contribution:** 2
**Rating:** 5
**Confidence:** 4

**Summary:**

This paper tackles the problem of catastrophic forgetting in continual learning, highlighting the limitations of traditional rehearsal buffer methods in data-sensitive contexts. The authors introduce HyperAdapter that employs hypernetworks to generate task-specific adapters for pre-trained models, thereby requiring fewer additional parameters as the number of tasks increases and promoting positive knowledge transfer across tasks.  Comprehensive experiments demonstrate that HyperAdapter consistently outperforms existing methods on benchmarks.

**Strengths:**

- The proposed HyperAdapter leverages hypernetworks to generate task-specific adapters for pre-trained models, addressing data privacy concerns and enabling effective knowledge transfer.
- HyperAdapter requires fewer additional parameters as the number of tasks increases, making it suitable for long-sequence continual learning.
- Experiments demonstrate that HyperAdapter consistently outperforms some methods in rehearsal-free continual learning.
- The paper is clearly written and easy to follow.

**Weaknesses:**

- The core idea proposed in this paper, using hypernetworks to generate model parameters (whether for all parameters, some parameters, or even applied to pre-trained models) to tackle the continual learning problem, has already been extensively explored in the literature [1-6]. This paper merely applies these existing methods in the context of prompting-based continual learning with pre-trained models, which significantly limits its novelty and contribution.
- Several of the innovative designs introduced, such as block-wise hyper-adapters, bear strong similarities in motivation and methodology to chunk embeddings and network partitioning discussed in [1]. This further constrains the novelty of the work.
- One of the claimed main advantages, "eliminating the necessity of knowing the task identities during inference," was previously addressed in [1] under the concept of unknown task identity inference. Additionally, the query-key matching mechanism commonly used in prompt-based continual learning to address this issue is a well-established practice [7-9].

[1] Continual learning with hypernetworks. ArXiv:1906.00695 2019.

[2] Continual learning with dependency preserving hypernetworks. Proceedings of the IEEE/CVF Winter Conference on Applications of Computer Vision 2023.

[3] Continual model-based reinforcement learning with hypernetworks. 2021 IEEE International Conference on Robotics and Automation.

[4] Hypernetworks for continual semi-supervised learning. ArXiv:2110.01856 2021.

[5] Partial hypernetworks for continual learning. Conference on Lifelong Learning Agents 2023.

[6] Prototype-based HyperAdapter for Sample-Efficient Multi-task Tuning. Proceedings of the 2023 Conference on Empirical Methods in Natural Language Processing.

[7] Bridging pre-trained models to continual learning: A hypernetwork based framework with parameter-efficient fine-tuning techniques. Information Sciences, 2024.

[8] Learning to Prompt for Continual Learning. CVPR 2022.

[9] DualPrompt: Complementary Prompting for Rehearsal-free Continual Learning. ECCV 2022.

**Questions:**

- What fundamental differences exist between hyperadapters and existing methods that utilize hypernetworks for continual learning [1-6], aside from variations in application scenarios? Please clarify the key innovations.
-  Is it possible to extend this approach to other continual learning settings, such as class-incremental, domain-incremental, or task-incremental learning?

---

### Official Review · Reviewer_BBsP · 2024-11-05

**Soundness:** 3
**Presentation:** 3
**Contribution:** 3
**Rating:** 6
**Confidence:** 4

**Summary:**

The paper presents a novel rehearsal-free approach for continual learning based on hypernetworks that generate so-called adapters to adapt the pre-trained model to different tasks.
The idea is intuitive and at high level is brain-inspired:
- the task dictionary is sort of an episodic memory in the hippocampus
- the hypernetwork is sort of the neocortex, storing past knowledge.
- task-specific embeddings are updated rapidly; and the general hypernetwork is updated slowly
The empirical results on the introduced by the authors CL-100 benchmark look promising.

**Strengths:**

- An intuitive high-level connection to the Complementary Learning Systems theory.
- A simple approach with promising performance wrt accuracy.
- Promising approach wrt scalability - HyperAdapter allows to avoid the excessive number of adapters.
- Overall, the paper is well-written and is comprehendable to a wide audience.

**Weaknesses:**

- It is not fully clear how the technical novelty is positioned, as well as what baselines should be included for demonstrating what ideas actually work (e.g. other hypernetworks-basd CL? other rehearsal-free CL approaches?).
- Overall, the technical novelty is rather modest: hypernetworks have been used for CL in the past. The ideas of exploring a combination of faster and slower adapting models have been explored in the past. The idea of recognizing already observed/recurrent tasks/concepts have been studies in the past too. (however, the studied combination of ideas in the context of adapting pre-trained models is novel to the best of my knowledge).
- The code and the experimental workbench is not available yet. Hence it is not easy to reproduce the results.

**Questions:**

- what are the closest existing approaches to the proposed HyperAdapters, only multiple adapters?
- what are the goals of the experiments?
- is it established that CL-100 is a better benchmark / good enough to be able to obtain conclusive empirical evidence (wrt the main goal of the experiments)?
- why the hyperparameters in Equations 3 and 10 are both set to 0.1
- how good/representative the initial pre-trained model should be for the main conclusions to hold?
- what is the expected efficiency gain compared to learning more adapters? does it depend on how similar/dissimilar different tasks are?

---

### Note · Authors · 2024-11-15

I have read and agree with the venue's withdrawal policy on behalf of myself and my co-authors.